# Exploring Artificial Neural Networks Efficiency in Tiny Wearable Devices for Human Activity Recognition

**DOI:** 10.3390/s22072637

**Published:** 2022-03-29

**Authors:** Emanuele Lattanzi, Matteo Donati, Valerio Freschi

**Affiliations:** Department of Pure and Applied Sciences, University of Urbino Piazza della Repubblica 13, 61029 Urbino, Italy; m.donati20@campus.uniurb.it (M.D.); valerio.freschi@uniurb.it (V.F.)

**Keywords:** artificial neural networks, human activity recognition, wearable devices, machine learning

## Abstract

The increasing diffusion of tiny wearable devices and, at the same time, the advent of machine learning techniques that can perform sophisticated inference, represent a valuable opportunity for the development of pervasive computing applications. Moreover, pushing inference on edge devices can in principle improve application responsiveness, reduce energy consumption and mitigate privacy and security issues. However, devices with small size and low-power consumption and factor form, like those dedicated to wearable platforms, pose strict computational, memory, and energy requirements which result in challenging issues to be addressed by designers. The main purpose of this study is to empirically explore this trade-off through the characterization of memory usage, energy consumption, and execution time needed by different types of neural networks (namely multilayer and convolutional neural networks) trained for human activity recognition on board of a typical low-power wearable device.Through extensive experimental results, obtained on a public human activity recognition dataset, we derive Pareto curves that demonstrate the possibility of achieving a 4× reduction in memory usage and a 36× reduction in energy consumption, at fixed accuracy levels, for a multilayer Perceptron network with respect to more sophisticated convolution network models.

## 1. Introduction

The continuous development of technology has made possible the production of electronic wearable systems equipped with sensing, computation, and communication capabilities with reduced power consumption, size, and cost. This advancement represents a key enabler for pervasive computing applications, within the framework of the Internet of Things (IoT). Indeed, ubiquitous sensing of various physical quantities allows to continuously monitor human activity, from which knowledge can be distilled. Human activity recognition (hereafter also denoted as HAR) consists of the recognition of typical activity patterns from signals gathered from sensing devices. This can be of particular interest in several fields such, for instance, health or surveillance applications [1,2].

HAR techniques are, according to a widely accepted coarse grain categorization, usually distinguished into video-based and sensor-based. The former entails the collection of video or image data from specific devices, whereas the latter is based on motion data signals gathered from inertial measurement unit (IMU) sensors (e.g., accelerometer, gyroscope), or from other information sources (e.g., microphone or Bluetooth transceiver). Clearly, sensor-based HAR represents a more flexible and cheap solution for continuous monitoring, while being more suitable in terms of privacy issues.

Machine learning algorithms are typically applied for the recognition of activity patterns. Classical techniques (e.g., decision trees, Support Vector Machines, Hidden Markov Models) represent a useful approach, but often require careful feature extraction and engineering. In this sense, neural networks (NN) provide an appealing alternative for designing end-to-end learning systems with high-performance levels, directly from raw sensor data. In particular, deep neural networks (DNN, for short) represent nowadays the state of the art in many machine learning applications, from computer vision to speech recognition tasks. The price to pay to achieve the impressive performance of DNN is the computational burden posed by training and inference activities. As a matter of fact, an option for overcoming computational issues is represented by resorting to GPU/TPU powered, often cloud-based, resources, which is particularly suitable for model training, according to a “once and for all” paradigm. On the other hand, shifting inference towards edge nodes of an IoT system provides several appealing benefits: *(i)* it avoids latency issues due to communications to and from the cloud, resulting in improved responsiveness; *(ii)* it enables higher levels of privacy and security by keeping most of the data on remote devices; *(iii)* it can improve energy efficiency by trading off computation and communication energy requirements.

Given the wide diffusion of tiny, low-cost, resource-constrained microcontroller units (also denoted as MCUs) in disparate IoT devices, the possibility of performing nontrivial inference directly on top of such devices is deemed to be of remarkable importance. In fact, memory and computation limits posed by most common general-purpose embedded MCUs impose tight constraints on the type and performance levels of machine learning tasks (e.g., human activity recognition) that could be executed.

Departing from these premises, we propose in this work to investigate the efficiency of two neural network models, namely a multilayer Perceptron (MLP) and a convolutional neural network (CNN). The proposed analysis is aimed at characterizing this efficiency in terms of CPU cycles, used memory, and energy consumption of the models, when a typical HAR task has to be executed on low-power MCUs on top of a real wearable device.

Within this framework, the main contributions of our study can be summarized as follows: *(i)* we propose an experimental workflow to analyze the performance of different machine learning models implemented on resource-constrained devices targeting HAR applications; *(ii)* we experimentally characterize on a wearable platform the trade-off between energy consumption or memory usage and classification accuracy for two types of widely used machine learning models; *(iii)* we explore several points of the design space (e.g., the latency of inference for different time windows of the acquired signals and feature selection) to provide a thorough assessment of the different models. The results of this investigation allow drawing some indications to potentially support academic and industrial research design of machine learning solutions to be deployed on edge computing devices.

The remainder of the article is structured as follows: in Section 2 we overview some state of the art works in the scientific literature related to our contribution; in Section 3 we describe the proposed methodology; in Section 4 we illustrate the setup adopted to conduct experiments, whose results and findings are provided Section 5; in Section 6 we conclude with some final remarks and observations.

## 2. Related Works

The idea of combining machine learning (ML) and IoT is an emerging research area that can be explored in different ways. A first distinction can be made based on where ML algorithms are placed (i.e., on the cloud or on the edge). Running ML in the cloud, for instance, offers potentially unlimited computational and storage resources at the cost of transferring data through the network [3,4,5]. On the other hand, moving ML from the cloud to the edge requires bringing complex algorithms to embedded devices characterized by evident resources constraints in terms of energy, storage, and computational power [6,7,8]. Moreover, some hybrid approaches have been proposed studying the suitability of partitioning heavy ML models on several sub-models which can be executed in a distributed manner across different IoT nodes or across nodes and remote clouds [9,10,11,12].

Focusing on the ML on top of edge computing (also referred to as TinyML), a reduced number of scientific contributions can be currently counted. In particular, TinyML refers to those platforms based MCUs with less than 1 MB of RAM and with extremely low energy consumption (about 1mW). In this context, most of the studies are, above all, aimed at demonstrating the possibility of executing the ML inference phase at the edge relying on centralized servers or distributed machines for training the models [13]. Porting ML inference on tiny devices comprises diverse optimization strategies ranging from reducing data input size by choosing more informative features [14] to lower the signal sampling frequency [15], and to reduce the number of bits that are used to represent a number to decrease the memory footprint [16,17]. A recent study presented by Feedorov in 2019 resumed most of these strategies and proposed an automatic tool to design Convolutional Neural Networks models suitable for deployment on MCUs used to classify low-size images [18].

Another dimension that can be used to classify TinyML IoT is represented by the type of learning tool being used. In particular, several contributions can be found in literature aimed at porting common machine learning models, such as Naive Bayes, Support Vector Machine, Decision Trees, or K-Nearest Neighbor, on top of IoT devices [19,20,21]. On the other hand, a number of recent studies also dealt with the optimization of deep learning models for tiny devices. For instance, Wang et al., proposed a tool to design an energy-efficient multilayer Perceptron for microcontrollers [22]. Lane et al., report a deep characterization of a CNN for resource constraint devices [23], while Disabato et al. [24] present a deep investigation about the accuracy of a CNN model tailored to two application scenarios (i.e., image classification and speech command recognition) on top of a Raspberry Pi and of an ARM Cortex-M7 MCU. Similarly, Wang et al., recently proposed a new run-time model compression method to deploy image classification DNNs on MCUs [25].

Although there are many works that study machine learning on tiny devices, to the best of our knowledge only few authors investigate also its effectiveness from an energy point of view and no contribution can be found for what concerns human activity recognition on wearable devices. In particular, the work by Feedorov in 2019, which proposed an automatic tool to design CNNs, characterizes the performance achieved during image recognition and reports, as reference values, energy consumption in the range from about 20 mJ to almost 400 mJ and an inference latency from about 30 ms to 500 ms depending on the dataset used [18]. In 2021, Odema et al., analyzed the trade-off between energy consumption and machine learning accuracy for low-power embedded devices targeted to detect a myocardial infarction [26] while Rashid et al., proposed an energy-aware online human eating activity recognition for wearable devices. In this case, the authors propose a hybrid approach that periodically updates the trained model over the network in order to meet the changes in people’s eating habits [27].

Finally, in the field of human daily activity recognition, Abdel et al., investigated the encoding of data, captured from heterogeneous sensors, into image representation to treat the task as an image classification problem. Authors test their proposed approach on two different public datasets but they do not provide any information about energy consumption and inference latency [28]. Novac et al., in 2020, in a preliminary work, compare supervised and unsupervised learning approaches deployed into an embedded device by analyzing the classification accuracy with respect to the ROM footprint and to the inference time. In this work, authors focus only on the inference time without considering time and energy spent to real-time collecting data from IMU sensors or to preprocess signals [29]. Recently, Alessandrini et al., presented a recurrent neural network (RNN), deployed on an embedded device, which takes in input data from Photoplethysmography (PPG) and tri-axial accelerometer sensors to infer the current human activity [30]. Similarly, Coelho et al. [17] and Mayer et al. [31] showed the adequacy of different deep learning models to be run on low-power platforms. Although the HAR on top of low-power devices is the common thread of these recent works, in none of the cases do authors analyze trade-offs between network complexity and resources utilization nor the impact on inference time or on measured energy consumption in real conditions.

On the other hand, in this work we conduct a novel accurate analysis aimed at characterizing artificial neural networks models efficiency in terms of CPU cycles, used memory, and energy consumption, when a typical HAR task must be executed on low-power MCUs on top of a real wearable device.

## 3. The Proposed Method

In this section, we describe the proposed methodology aimed at exploring the efficiency of artificial neural networks by means of a fine-tuning and characterization of the network complexity devoted to reduce the model computation needs in terms of CPU cycles, memory usage, and energy consumption without introducing an appreciable loss of accuracy. For this purpose, we chose a MLP and a CNN traditionally used in several classification problems. In particular, the MLP network represents a classical machine learning tool used by now for several decades while the CNN is a more recent tool which belongs to the deep learning category and which proved a very interesting classification performance and versatility. Indeed, CNNs have been demonstrated particularly suitable for HAR tasks (see for instance [1,2]); we chose to also include MLP in our study because they represent a well-grounded type of machine learning model that could potentially pose a lighter burden on resource-constrained devices. Interestingly, the experimental results show that similar performance (in terms of classification accuracy metrics) can be reached for the application under study for both machine learning models, but with different energy consumption, memory usage, and inference latency, making it reasonable to take into consideration the implementation of MLP within IoT edge computing HAR applications. The two networks have been trained and tested using, as a case study, the recognition of daily human activities, and as the wearable device a real development platform.

### 3.1. Perceptron Network

A Perceptron network is typically made up of a set of sensory units representing the input layer, one or more hidden layers, which are made up of computation nodes, and one output layer, which computes the final result. A network with only one hidden layer is known as a Single Layer Perceptron (SLP), whereas a network with multiple hidden layers is known as a Multi-Layer Perceptron (MLP). Because the input signals in both cases propagate forward through the network, they are also known as feedforward networks. Each neuron receives input signals, processes them, and transmits the result to the neurons to which it is connected. The processing involves evaluating a transfer function on the weighted sum of the received inputs, with each weight representing the effectiveness of a synaptic connection (i.e., of an input line). The neural network’s learning capabilities are achieved by adjusting the weights in accordance with the learning algorithm of choice. Various training algorithms and performance metrics have been proposed in the literature [32]. Before sending data to the network, data must be preprocessed by means of a processing phase called feature extraction. This phase entails incorporating domain knowledge into data processing in order to reduce its complexity and generate summary descriptors that improve the performance of learning algorithms. Feature processing takes time and requires specialized knowledge because the features vary depending on the problem being analyzed.

### 3.2. Convolutional Neural Network

A CNN is essentially a type of MLP. It takes a novel approach that makes use of any spatial or temporal information in the data. In fact, the CNN was inspired by the biological process that occurs in the animal visual cortex, where neurons only handle responses from specific regions of the visual field. Convolving filters are used by CNNs to handle local regions within the data, emulating the visual cortex. The network structure is mainly composed of an input layer, convolutional layers, pooling layers, and fully connected layers. The input layer is responsible for collecting data and forwarding it to the next layer. The convolutional layer, which contains several convolution filters (kernels) that convolve with the input data, is the main core of a CNN and it is responsible for reducing the dimension of the input data automatically extracting useful features. The pooling layer, also known as the subsampling layer, is then used to further reduce the number of parameters and the resulting computational cost by incorporating max-pooling and average-pooling operations. Finally, a fully connected layer functions as a traditional Perceptron network, taking in input from the previous layer’s features. A CNN has traditionally been used in the Deep-Learning approach due to its ability to eliminate the need for feature extraction and feature selection, often at the expense of increased computational complexity and memory usage [33].

### 3.3. Tuning Neural Networks Complexity

Artificial neural networks are complex systems whose performance and computational needs depend on several parameters. These parameters can be divided into *hyperparameters* and *learnable parameters*.

The term hyperparameters refers to those parameters that are used to control the network structure and the learning process so that they can be divided, in turn, into *structural hyperparameters* and *algorithmic hyperparameters* [34]. Structural hyperparameters, also called model hyperparameters, describe the network structure and topology and are represented, for instance, by the number of layers, the number of neurons in each layer, the degree of connectivity, the neuron transfer function, etc. As they directly modify the structure of the network, they affect both its effectiveness, its computational complexity, and its memory footprint. On the other hand, algorithmic parameters are the training algorithm, the learning rate, the momentum, the training set size, and so on, and they are used to control the learning process. These parameters are not part of the model and they have no influence on its performance but they affect the speed and the quality of the learning process [34].

Finally, learnable parameters, also referred to as trainable parameters, are represented by weights and biases of the neuron connections and are modified during the learning phase by the training algorithms. The number of these parameters strictly depends on the network type and on the choice of the structural hyperparameters and it straight influences the computational complexity and memory footprint of the model.

Definitely, tuning a neural network entails tuning its parameters following a certain objective function. Traditionally, network-tuning is done to optimize its performance in order to obtain the highest effectiveness in terms of accuracy [35,36,37,38]. In this work, on the contrary, we aim at tuning the network complexity to reduce the model computation needs (CPU cycles, memory usage, and energy consumption) to meet the resources constraints of wearable devices, and, at the same time, to minimize the loss in the model performance. For this purpose, we train the network by means of a traditional desktop computer, without computation constraints, and then install the trained model into the wearable device to infer a real-time classification. Thanks to this solution we can leave the training phase as complex as we need to obtain the high accuracy, and tune only the model computation needs. In other words, we focus on the tuning of the structural hyperparameters and of the number of the trainable parameters to obtain a good trade-off between inference accuracy and resources utilization.

In particular, in this work, we study how the tuning of the complexity of a Perceptron and of a CNN network, by properly setting its hyperparameters, influences its performances and its suitability to be installed and run on a wearable device. For this purpose, we use a real smartwatch, namely Hexiwear [39], on top of which we install the two neural network models used to classify daily human activities.

### 3.4. The Case Study

HAR refers to the task of automatically recognizing daily activities carried out by an individual. In particular, human activities such as walking, running, sitting, standing, driving, sleeping, etc. are detected and classified by means of data collected from several sources, ranging from inertial wearable sensors (e.g., accelerometer and gyroscope) to video capture devices (e.g., visible or infrared images and streams).

Particularly, inertial sensors have been employed in HAR systems for health-related applications targeting, for instance, remote monitoring of elderly people, fall detection, medical diagnosis, physical therapy for rehabilitation, and biomechanics research. Inertial sensors have been also adopted for gait recognition approaches, with the aim of recognizing users by their walking patterns, through motion signals collected from accelerometers and gyroscopes mounted on board of wearable devices [40].

Despite a huge literature on the topic, there is not a common agreement on the number and types of sensors to be adopted, neither on their positioning nor on the methods most suitable for recognizing most common patterns.

Computer vision-based technologies have been proposed as well for HAR tasks, although with peculiar issues that typically constrain their applicability. In fact, video cameras or other types of external sensors are usually placed in predefined locations and the collection of useful data can be carried out only within the range of these sensors in order to interact with them.

Machine Learning (in particular deep learning) provides a valuable framework of techniques that have been recently used for HAR. Indeed, Machine Learning models allow building reliable and mathematically sound tools, that can take advantage of the increasing amount of information that can be collected from sensing devices worn by individuals. Motion signals can be continuously gathered, divided into chunks of predefined window size [41] and eventually processed and classified by means of suitable tools such, for instance, artificial neural networks.

Accurately recognizing human activity patterns such as running, walking, sitting, climbing or descending the stairs, jumping, lying down, poses several challenging issues. First of all, the accuracy of the classification process can be strongly affected by the position of sensors, hence their optimal placement is key to achieving correct predictions. The use of multiple inertial sensors located both in the upper and in the lower part of the body may improve performance, often at the cost of decreased usability. Other issues relate to the privacy and security of individuals, which prompt for keeping the inference phase on top of wearable devices. As a consequence, the inference must be performed *at the edge*, possibly with a reduced impact on energy consumption.

For these reasons, we focused in this study on the RealWorld HAR dataset described by Sztyler et al., in 2017 [42]. The data set contains the acceleration, GPS, gyroscope, light, magnetic field, and sound level data collected at a sampling frequency of 50 Hz of the following activities: climbing stairs down and up, jumping, lying, standing, sitting, running, and walking. Each activity was performed for ten minutes (except for jumping which was repeated for about 2 min) by fifteen subjects of different ages and gender in real-world conditions. Each subject was instrumented with different wearable sensors in order to capture signals generated in different body positions such as chest, forearm, head, shin, thigh, upper arm, and waist.

Since this work focuses on the classification performance of wearable devices such as smartwatches, we refer only to those signals commonly available on these devices (accelerometer and gyroscope) and collected in the subject’s forearm.

### 3.5. The Proposed Workflow

Figure 1 reports a graphical representation of the experimental workflow proposed in this study. In particular, after a first step involving a pre-processing phase of the raw data extracted from the case study dataset, the two classification networks (i.e., the Perceptron and the CNN networks) are trained and tested using a k-fold strategy. The resulting trained models are then optimized in order to reduce the memory footprint and the inference latency by means of the standard quantization scheme implemented in TensorFlow Lite [43]. The next step takes as input the optimized models and translates them into a C language representation that can be cross-compiled for the selected wearable platform. The output of this step consists of a binary file representing the firmware that is used to program the wearable device to continuously read signals from its onboard sensors and to execute the model to obtain the classification inference. Finally, the inference phase of the installed model is characterized in terms of memory usage, CPU cycles, and energy consumption by means of the measurement setup.

Notice that, the only two processing phases that need to meet the resources constraints of the wearable devices are respectively the pre-processing and inference phases. The first one must be replicated on the device to maintain consistency with the trained model input and, obviously, the second one is the actual phase of using the neural network to classify the HAR. All the remaining phases can be left as complex as we need to obtain the highest classification accuracy.

### 3.6. Signal Pre-Processing

The six signals extracted from the dataset (i.e., the 3-axial accelerometer and gyroscope data) have been divided into time windows and each of these has been considered as a sample to be used to train and to test the artificial neural networks.

In order to evaluate the classification performance of the Perceptron feed-forward network, starting from each time window, two sets of descriptive features have been computed. In particular, the first set contains basic statistical descriptors aimed at capturing data tendency and variability. These descriptors are: *(i)* average (A); *(ii)* standard deviation (S); *(iii)* maximum value (X); *(iv)* median value (M) while the second set is built with Kurtosis (K) and Skewness (W) parameters aimed at capturing the shape of the data. Kurtosis and Skewness, in fact, are used to describe, respectively the degree of dispersion and symmetry of the data. In particular, Kurtosis is a measure of whether the data are heavy-tailed or light-tailed relative to a normal distribution while, Skewness measures how much data differ from a completely symmetrical distribution [44].

In the case of the CNN network, on the other hand, no feature extraction is needed and the samples of the signals that make up the time window have directly been used as input for the network.

Obviously, the size of the time window influences the performance of classification models in different ways. First of all, it must be large enough to capture the “fingerprint” of the human activity in order to be properly recognized but it must not be too large to include consecutive activities. For what concerns HAR, different window lengths have been used in the literature: starting from 1 s up to 30 s [45,46,47,48,49]. Of course, the size of the window strictly depends also on the activities to be recognized and, moreover, it also affects other aspects such as the model size and the inference computation and energy costs. For these reasons in a later section, we report a deep sensitivity analysis of the classifiers with respect to this parameter.

## 4. Experimental Setup

### 4.1. The Software Platform

The software platform used to build the proposed workflow described in Section 3.5 is based on the Python open-source library for artificial neural networks called Keras [50] running on top of TensorFlow [51]. Then, the trained model has been optimized and converted into a C language representation by means of TensorFlow Lite framework [52]. Finally, the Arm Mbed development toolchain has been used to generate the executable image to be used to program the wearable device [53].

### 4.2. The Wearable Device

As a wearable device, we used a smartwatch called Hexiwear produced by MikroElektronika [39]. The main CPU is a low-power Arm Cortex-M4 from NXP running at 120 MHz and equipped with 1MB of non-volatile flash memory and with 256 KB of SRAM. The main CPU is connected with a triaxial accelerometer, magnetometer, and gyroscope for a total of 9 degrees of freedom (DoF), and with an OLED display with a resolution of 96 × 96 pixels. In addition, such a device has additional capable sensors to detect external temperature, atmospheric pressure, humidity level, and heart rate. The whole device is powered by means of a 190 mAh Li-Po battery that, thanks to the low-power consumption of the main CPU (about 35 mA at maximum speed), ensures a good execution time.

### 4.3. Energy Consumption Measurement Setup

In order to monitor the energy expenditure of the smartwatch, we measured the voltage drop across a sensing resistor (39 Ω) placed in series with the power supply of the device. The smartwatch was powered at 3.3 V through a NGMO2 Rohde & Schwarz dual-channel power supply [54], and we sampled the signals to be monitored during the experiments by means of a National Instruments NI-DAQmx PCI-6251 16-channel data acquisition board connected to a BNC-2120 shielded connector block [55,56].

### 4.4. Classification Performance Metrics

For both networks, we calculate several classification performance metrics, together with the standard deviations, during a k-fold cross-validation test with k = 5. In particular, dealing with multi-class classifiers, the following quantities have been evaluated for each of the eight classes (i∈[1⋯8] is an index that identifies a specific class): $TP_{i}$, the number of true positives predicted for class *i*; $TN_{i}$, the number of true negatives predicted for class *i*; $FP_{i}$, the number of false positives predicted for class *i*; $FN_{i}$, the number of false negatives predicted for class *i*.

Subsequently, these indicators have been used to compute the following metrics (corresponding to the so called *macro-averaging* measures) [57]:(1)$$Precision=\frac{1}{8}\sum_{i=1}^{8}\frac{TP_{i}}{TP_{i}+FP_{i}}$$
(2)$$Recall=\frac{1}{8}\sum_{i=1}^{8}\frac{TP_{i}}{TP_{i}+FN_{i}}$$
(3)$$F1score=2\cdot \frac{Precision\cdot Recall}{Precision+Recall}$$
(4)$$Accuracy=\frac{1}{8}\sum_{i=1}^{8}\frac{TP_{i}+TN_{i}}{TP_{i}+TN_{i}+FP_{i}+FN_{i}}$$

## 5. Experimental Results and Discussion

In this section, we report the extensive experiments conceived to assess the adequacy of the two representative network models to be used on top of a low-power wearable device for real-time inference of human activity. In particular, we explore the hyperparameters space of the two networks, and then we compare the two sets of results by taking into account the impact on the computation needs such as CPU cycles, memory usage, and energy consumption.

### 5.1. Perceptron Network

The performance of the Perceptron network has been investigated while changing: (*i*) the network structure and size, (*ii*) the size of the window processing, and (*iii*) the number of input features.

#### 5.1.1. Network Size and Structure

The size of a Perceptron network can be expressed in terms of the total amount of neurons contained in its hidden layers. In fact, for a given input (i.e., a constant number of input signals or features) and for a given classification task (i.e., a constant number of classes to be recognized) the size of input and output layers are statically defined while the size of the network can be changed by varying both the number of hidden layers and the number of hidden neurons in each layer.

Figure 2 shows the structures of a Single Layer Perceptron (SLP) and of a Multi-Layer Perceptron (MLP) with two hidden layers. The input size, the number of hidden neurons, and the output size are defined respectively by *m*, *n*, and *k* parameters. Notice that in the MLP, each of the two hidden layers contains n/2 neurons so that they can be compared with the SLP configuration. Moreover, before and after the fully connected layers a dropout layer (not shown in the figure) with a dropout rate set to 0.5 was inserted to help prevent overfitting. Finally, as the activation function, the standard ReLU was used.

In Table 1 the size of the two networks, expressed in terms of hidden neurons (*n*), together with the number of its trainable parameters are shown. Notice that, the number of trainable parameters is calculated for an input size (*m*) equal to 36 and for an output size (*k*) of 8. The input size results from the extraction of the 6 features, described in Section 3.6, from the 6 independent signals (i.e., data from the triaxial accelerometer and gyroscope) while the output size reflects the number of classes described in Section 3.4. For sake of clarity, each network configuration has also been identified by assigning a unique ID (first column of the table) Finally, the table also shows that, due to the fully-connected nature of the hidden layers, the number of trainable parameters in the MLP grows faster with respect to the SLP when increasing the total number of hidden neurons.

Figure 3 reports the classification performances metrics, together with the standard deviations, calculated during the k-fold cross-validation tests obtained by the SLP (a) and by the MLP (b) networks when varying the network size accordingly to Table 1. Notice that, the subscript used to index the network configurations reported on the abscissa axis represents the unique ID of Table 1 so that the network size increases from the left to the right of the graphs.

For both networks, the highest performances have been measured on the configuration containing the highest number of hidden neurons (i.e., $SLP_{5}$ and $MLP_{5}$) where SLP and MLP show, respectively, an average classification accuracy of about 84.7% and 86.9%.

Figure 4 reports two confusion matrices obtained during the k-fold cross-validation test, respectively, by $SLP_{5}$ (a) and by $MLP_{5}$ (b) network configurations. Although the average accuracy of the two classifiers differs only by about 2%, the confusion matrices point out that the MLP network obtains a more balanced precision between classes with respect to SLP. For instance, the “Climbing up” activity was misclassified as “walking” 147 times with SLP leading to a recall of about 70.7% while with the MLP classifier it grew up to 82.3%. Similarly, the MLP network was able to better discriminate between “Lying” and “Sitting” increasing the recall of the former of about 4.7 percentage points.

The networks trained during k-fold cross-validation experiments have then been optimized and translated into TensorFlow Lite models in order to be installed into the wearable device. Once installed, each model has been used to infer the activity classification starting from the data read from the device internal sensors. During this phase, the energy spent to carry out a single inference and the memory usage of each model have been measured. In particular, the measured inference energy includes the contributions related to the following phases: (i) sensor reading (accelerometer and gyroscope), (ii) feature extraction, (iii) model evaluation, and (iv) writing the output label (HAR class) on the device display. Notice that, as in the previous experiment it emerged that the MLP network family outperforms the SLP one, only the former has been characterized in terms of memory and energy needs.

Figure 5a,b report the Pareto curves plotting the trade-off between performance loss and, respectively, memory usage and energy consumption when varying the network size. In particular, the best trade-off between memory usage and performance loss comes from $MLP_{3}$ and $MLP_{4}$. In fact, going from $MLP_{4}$ to $MLP_{5}$, for instance, reduces the performance loss by only about 0.004 (i.e., 0.4%) at the cost of almost quadrupling the memory occupation which grows from about 22 KB to 74 KB. In the same way, Figure 5b shows that, from the energy consumption point of view, the best configurations are once again $MLP_{3}$ and $MLP_{4}$ which maintain the energy spent for a single inference close to 1 mJ, while for $MLP_{5}$ more than 2.4 mJ are needed.

Notice that, since configuration $MLP_{4}$ turned out to be the best trade-off between classification performance and resource utilization, succeeding experiments were performed exclusively using this configuration.

#### 5.1.2. Window Size

As described in Section 3.6 the size of the processing window influences the performance of the classification models in several ways. In this section, the results of the in-depth analysis of this dependence are reported. In particular, Figure 6a shows the classification metrics obtained by $MLP_{4}$ network configuration when varying the window size. Interestingly, for a processing window in a range from 2 to 6 s, the classification performances do not seem to have a recognizable trend while the shortest window (i.e., 1 s) produces the worst results.

Another key to interpretation, however, is shown in Figure 6b where the influence of the size of the processing window on the inference time and on the model size are reported. In particular, the inference time of the model linearly increases when the window size increases. This is due to the fact that increasing the window size increases the time, and consequently the energy, needed to extract the representative features. In particular, the inference time reported on this graph was obtained with a pre-processing phase entailing the calculation of the average (A), the standard deviation (S), and the maximum (X) values for each of the six input signals. Notice that, when the HAR is provided in real-time by a low-power wearable device, the inference latency must be carefully taken under control to be sure to meet the temporal deadlines which must not exceed the size of the processing window. In the current experiments, the deadline is widely met considering that for the larger processing window (i.e., 6 s) the inference time turns out to be about 19 milliseconds.

Finally, the memory footprint of the model is not affected by the size of the processing window as the number of neurons on the input layer only depends on the number of the calculated features. In order to maintain a good trade-off between classification performance and resource utilization, for all the experiments conducted in this study, a window of intermediate size (i.e., 3 s) has been chosen.

#### 5.1.3. Feature Selection

In order to evaluate the relative influence of the proposed features on the classification performances, we use the forward feature selection method [58]. Forward feature selection relies on an objective function (e.g., the accuracy) which is used as a criterion to evaluate the impact of adding a feature from a candidate subset, starting from an empty set until adding other features does not induce any improvement in the objective function. We applied this strategy to highlight how the features described in Section 3.6 contribute to the overall performance of the classifier.

Table 2 shows the classification performances when varying the adopted features. For each performance metric, the maximum value achieved has been highlighted in bold.

All metrics showed a monotone increasing trend until adding the first three features (i.e., Average, Standard deviation, and Maximum value). Then, adding more features does not involve a further increase in measured performances. This demonstrates that not all features provide original information content useful for the classification process.

Obviously, extracting more features also increases the computational and energy costs of the pre-processing phase which, moreover, depends in a nonlinear manner on the selected features. In order to guarantee the best trade-off between classification performance and computational complexity, in all the experiments reported in this study, the average value (A), the standard deviation (S), and the maximum value (X) have been selected.

### 5.2. Convolutional Neural Network

The performances of the CNN have been evaluated by varying its network structure and size and by changing the size of the input window.

#### 5.2.1. Network Size and Structure

A CNN used in classification tasks is generally composed of an input layer receiving the raw signals, followed by several convolutional layers each of which is followed by a pooling layer. In this study, we use convolutional 1D layers provided by TensorFlow to convolve the input signal over the temporal dimension. In particular, we provide to the input layer the six selected signals as vectors with a number of timesteps which depends on the size of the selected input window. Downstream of the convolutional layers, a Flatten layer distributes the convolved signals to one or more fully-connected layers which are used to properly act as a classifier. Finally, the network terminates with a standard output layer.

In this study, we investigated two families of CNNs (the structure of which is shown in Figure 7) that we have identified with the labels CNN_1.1 and CNN_2.2. The family indexed by _1.1 contains a single convolutional layer followed by a max-pooling layer and by a single fully-connected layer while, the family indexed by _2.2 has two convolutional, two max-pooling and two fully-connected layers. The size of each network instance is then defined in terms of the number of convolutional filters (*j*), hidden neurons (*n*), and output classes (*k*) while the size of the max-pooling layer is constant (size = 3).

Before and after the fully-connected layers, a dropout layer (not shown in the figure) with a dropout rate set to 0.5 was inserted to help prevent overfitting. In each layer, the standard ReLU was used as the activation function.

In Table 3 the size of the two CNN families, together with the number of its trainable parameters, have been reported. For CNN_2.2 network, which contains two fully-connected layers, the number of hidden neurons reported on the table is the sum of the neurons of the two layers. Interestingly, in CNN_1.1 network the number of trainable parameters grows faster with respect to CNN_2.2 when increasing the total number of hidden neurons. This is due to the lack of downsampling activity of the input representation achieved by the single couple of convolutional and pooling layers of CNN_1.1. Each network configuration, identified by the assigned unique ID (first column of the table), has been fully tested according to the proposed workflow.

Figure 8 reports the classification performances metrics, together with the standard deviations, calculated during the k-fold cross-validation tests obtained by CNN_1.1 (a) and by CNN_2.2 (b) networks when varying the network size. For both families, the highest performances have been recorded with the configuration containing the highest number of hidden neurons. In this case, CNN_1.1 and CNN_2.2 show, respectively, an average classification accuracy of about 84.2% and 87.7%.

Figure 9 shows two confusion matrices obtained respectively by $CNN\_1.1_{5}$ (a) and by $CNN\_2.2_{5}$ (b) network configurations. As for the comparison between SLP and MLP networks, the confusion matrices point out that the network containing two fully-connected layers shows a more balanced precision between classes with respect to the single layer. For instance, similar classes such as ”Climbing up” and “Walking” are better discriminated in $CNN\_2.2_{5}$ where classification recall reaches 85.1% and 91.1% for ”Climbing up” and “Walking” classes respectively.

Notice that, as CNN_2.2 family outperforms CNN_1.1 in terms of classification accuracy, only the former has been optimized and translated into TensorFlow Lite models in order to be installed into the wearable device. Once installed, the inference phase of each model has been characterized regarding its energy consumption and memory usage. In particular, for CNNs the inference does not entail features extraction so that the only relevant contributions to the energy consumption are related to the following three phases: (i) sensor reading (accelerometer and gyroscope), (ii) model evaluation, and (iii) writing the output label (HAR class) on the device display.

Figure 10a,b report the Pareto curves plotting the trade-off between performance loss and, respectively, memory usage and energy consumption when varying the network size. In particular, the best trade-off, both in terms of memory usage and in terms of energy consumption with respect to the performance loss, is reached by $CNN\_2.2_{3}$ and $CNN\_2.2_{4}$. In fact, further increasing the network complexity up to $CNN\_2.2_{5}$ configuration, for instance, reduces the performance loss of about 0.8% at the cost of almost tripling both the memory occupation, which grows from about 91 KB to 338 KB and the energy consumption, which from about 39 mJ reaches about 125 mJ.

Notice that, since configuration $CNN\_2.2_{4}$ turned out to be the best trade-off between classification performance and resource utilization, succeeding experiments were performed exclusively using this configuration.

#### 5.2.2. Window Size

In this section, the results of an in-depth analysis of the dependence of the model performances when varying the size of the input window are reported. In particular, Figure 11a reports the classification metrics obtained by $CNN\_2.2_{4}$. The average values show a slight improvement when the window size increases even if, for larger values, there is also an increase in the corresponding standard deviations which reveal a greater instability of the results.

Figure 11b shows the influence of the size of the input window on the inference time and on the model size. Thanks to the fact that for a larger input window a larger convolutional layer is required, both the inference time and the model size linearly increase with the window size. So that, for instance, when the processing window increases up to 6 s the model size becomes about 170 KB and the inference time exceeds about 750 milliseconds. This suggests that for a CNN targeted for low-power embedded systems the size of the input window must be further carefully chosen not to impair the effectiveness of the system.

Even for CNN networks experiments conducted in this study, we have chosen an intermediate size (i.e., 3 s) in order to obtain a good trade-off between classification performances and resources utilization.

### 5.3. Network Comparison

After having explored the space of the hyperparameters to characterize the behavior of the two types of artificial neural networks, a direct comparison between these has been performed. First of all, we conducted a set of experiments to statistically assess whether the accuracy of the two families of classification models is comparable or not. In particular, a McNemar test with the confidence level of 95% has been performed for each couple of MLP and CNN_2.2 models to test the null hypothesis that the two classifiers have equal accuracy for predicting the true classes [59]. Table 4 reports the results of five runs of this test. In particular, for each run, the logical value H0, which represents the decision when testing the null hypothesis, together with the *p* value are reported. Notice that, a false value of the decision indicates that the null hypothesis is not rejected with a confidence level of 95% and this implies that the two classifiers statistically agree in the same way in classifying the results.

Each column of the table shows the results obtained when comparing the particular configuration of the two networks denoted by the subscript ID so that, for instance, the column $MLP_{1}-CNN\_2.2_{1}$ reports the results obtained when comparing the MLP containing 32 hidden neurons with the CNN consisting of two convolutional, two polling, and two fully-connected layers containing, in turn, 32 hidden neurons. The results show that only the last two combinations, namely $MLP_{4}$ versus $CNN\_2.2_{4}$ and $MLP_{5}$ versus $CNN\_2.2_{5}$, which report four false values out of five, statistically agree in the classification and can be considered interchangeable from a performance point of view. Starting from this consideration we can argue, for instance, that the choice of whether to use a $MLP_{4}$ or a $CNN\_2.2_{4}$ network must depend on the way in which they use the system resources, such as memory and energy, rather than from the classification performances.

Figure 12a,b show the Pareto curves of the accuracy loss plotted versus the memory usage and versus the energy consumption respectively of MLP and of CNN_2.2 network families.

From the memory usage point of view, the MLP network sharply outperforms CNN_2.2 in all configurations. For instance, $MLP_{4}$ uses about 22 KB of memory while $CNN\_2.2_{4}$ needs more than 90 KB. Moreover, $MLP_{5}$ and $CNN\_2.2_{5}$ are even further away using, respectively, about 76 KB and about 340 KB. Definitely, from the comparison of the configurations reaching the best trade-off (i.e., $MLP_{4}$ and $CNN\_2.2_{4}$), it results in an advantage of about 4x in favor of the MLP network.

Moreover, considering the energy consumption, the difference between the two networks is even more marked with a strong advantage in favor of the MLP network. In fact, the maximum energy consumed by the MLP network, charged to configuration $MLP_{5}$, is close to the minimum energy consumed by the lower complex CNN network (i.e., $CNN\_2.2_{1}$). Finally, comparing the energy consumption of the configurations reaching the best trade-off for the two types of network (i.e., $MLP_{4}$ and $CNN\_2.2_{4}$) we get an advantage of about 36x in favor of the MLP network.

Summing up, for a tiny device such as a smartwatch used to continuously recognize human activities, given an identical level of classification performance and considering an optimization criterion based on system resources such as memory and energy, it is convenient to use classic Perceptron networks rather than more complex deep convolutional neural networks.

## 6. Conclusions, Limitations, and Future Research

In this work we have studied the resource cost–accuracy trade off on different artificial neural networks targeted to the real-time classification of daily human activities by means of tiny devices. This is motivated by the challenge of improving the feasibility of these low-power wearable devices to perform inference at the edge, contributing to a more human-centric IoT. To this purpose, we empirically investigated how the resources needs and the classification accuracy depend on design choices and on network hyperparameters.

In the presented case of study, based on a public HAR dataset, we demonstrated that given an identical level of classification performance, it is more convenient, from the resource utilization point of view, to use classic Perceptron networks rather than more complex networks such as convolutional ones. In particular, Pareto curves reporting resource utilization Vs accuracy loss show a 4x advantage of the Multilayer Perceptron network for what concerns the memory usage, and of about 36x considering the energy consumption.

The major focus of this investigation was the characterization of two reference neural network architectures; while the conducted experiments allow drawing some conclusions to support systems development for this type of applications, several research directions remain open to overcome the limitations of the proposed study. First of all we used a single hardware as a reference platform for wearable devices while exploring other alternative options could be considered in order to further augment the spectrum of the results. Finally, extensive fine-tuning of the two studied networks by means of automatic neural architecture search methods could improve the exploration of the design space. Moreover, other machine learning models could be taken into account such, for instance, recurrent neural networks or other models suitable for processing sequential data therefore enabling a principled exploration of other points in the design space. Extending the proposed workflow methodology to other models clearly represents a challenge because of the computational burden placed by the search space and by the related training of models but would widen the findings of this work.

As an imminent future direction, we are planning to apply the *early exit technique* to more accurate deep CNNs in order to dynamically reduce its computation complexity and execute the inference, when energetically convenient, on the device while maintaining the possibility of exploiting the complete CNN classification power on the gateway or directly in the cloud.

In summary, our study confirms the multifaceted nature of the problem of executing human activity recognition tasks on board of edge-computing devices characterized by constrained resources; careful investigation of the complex interplay between hardware and software components can provide useful hints towards designing solutions compatible with strict energy requirements without significantly affecting accuracy levels.

## Figures and Tables

**Figure 1 sensors-22-02637-f001:**
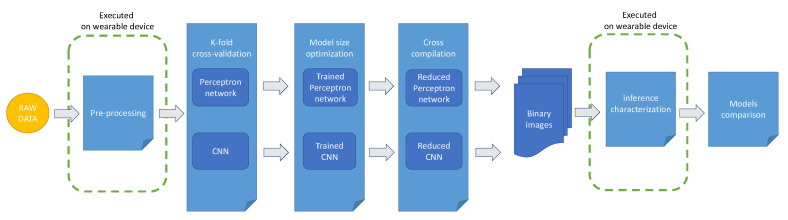
The proposed experimental workflow.

**Figure 2 sensors-22-02637-f002:**
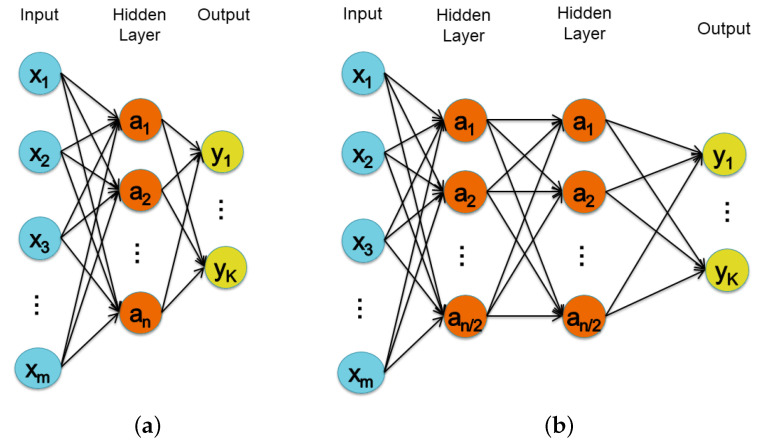
Structure of the SLP (**a**) and of the MLP (**b**) families network.

**Figure 3 sensors-22-02637-f003:**
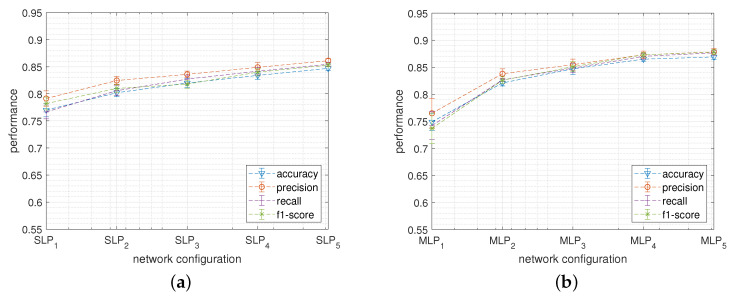
Classification performance obtained when varying the structure of the *SLP* (**a**) and of the *MLP* (**b**) networks.

**Figure 4 sensors-22-02637-f004:**
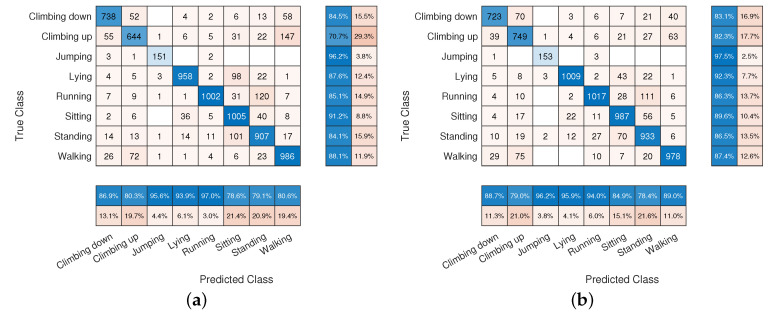
Confusion matrices obtained by means of the $SLP_{5}$ (**a**) and of the $MLP_{5}$ (**b**) networks.

**Figure 5 sensors-22-02637-f005:**
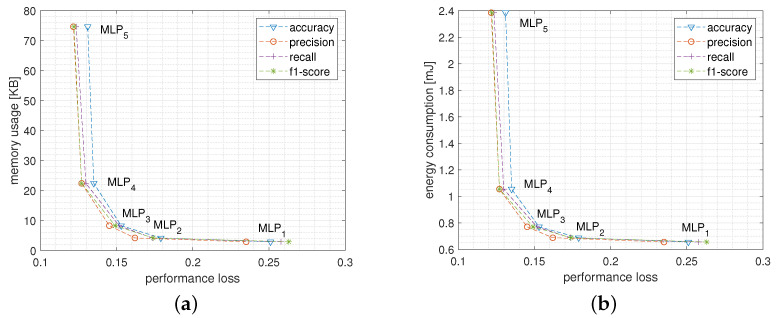
Pareto curves reporting performance loss versus memory usage (**a**) and versus energy consumption (**b**) of the MLP model when varying the network structure.

**Figure 6 sensors-22-02637-f006:**
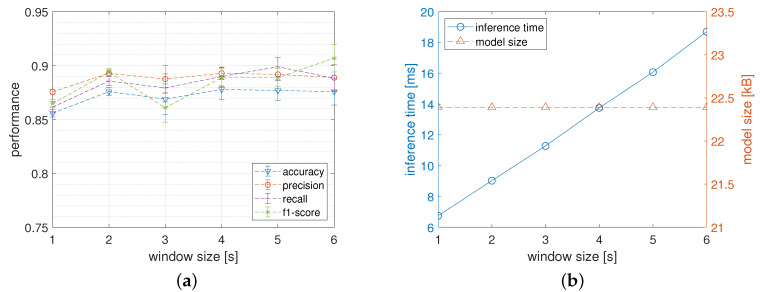
Network performance obtained by the $MLP_{4}$ network when varying the size of the processing window. Figure (**a**) reports the classification performance while Figure (**b**) plots the memory footprint and the inference time of the model calculated on the wearable device.

**Figure 7 sensors-22-02637-f007:**
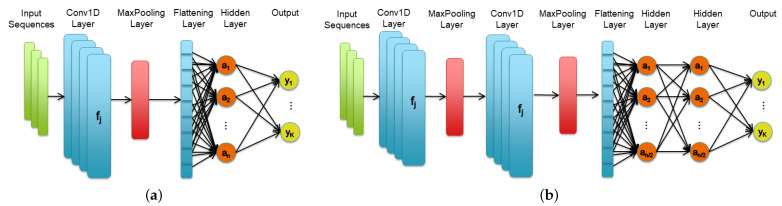
Structure of the $CNN_{1.1}$ (**a**) and of the $CNN_{2.2}$ (**b**) network families. The size of each network instance is defined in terms of number of convolutional filters (*j*), hidden neurons (*n*), and output classes (*k*) while the size of the max pooling layer is constant (size = 3).

**Figure 8 sensors-22-02637-f008:**
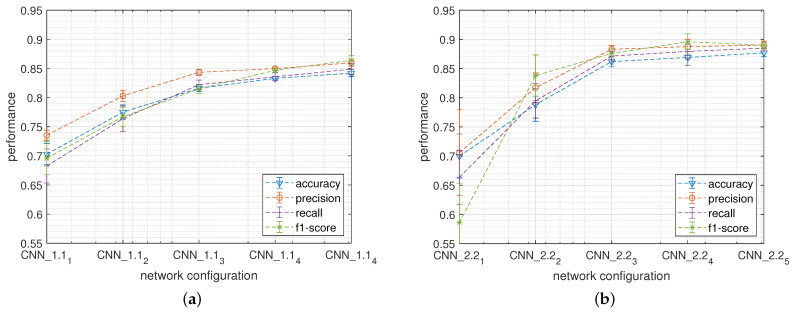
Classification performance obtained when varying the structure of the CNN_1.1 (**a**) and of the CNN_2.2 (**b**) networks.

**Figure 9 sensors-22-02637-f009:**
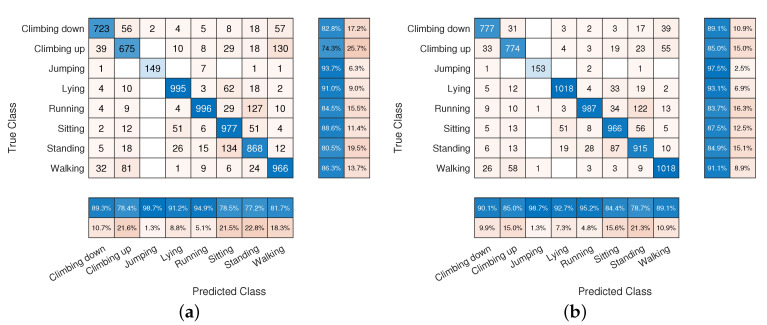
Confusion matrices obtained by means of the $CNN\_1.1_{5}$ (**a**) and of the $CNN\_2.2_{5}$ (**b**) networks.

**Figure 10 sensors-22-02637-f010:**
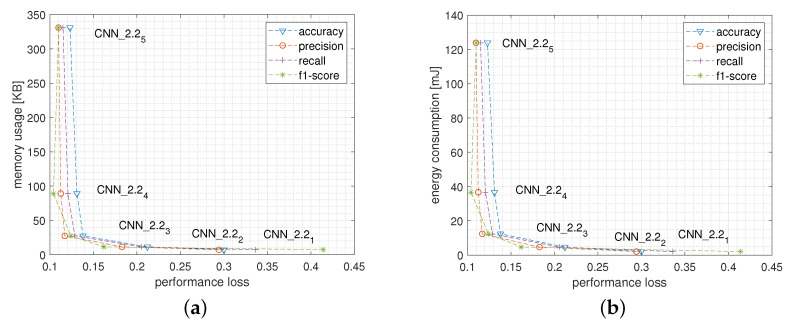
Pareto curves reporting performance loss versus memory usage (**a**) and versus energy consumption (**b**) of the CNN_2.2 model for a single inference executed on the wearable device when varying the network structure.

**Figure 11 sensors-22-02637-f011:**
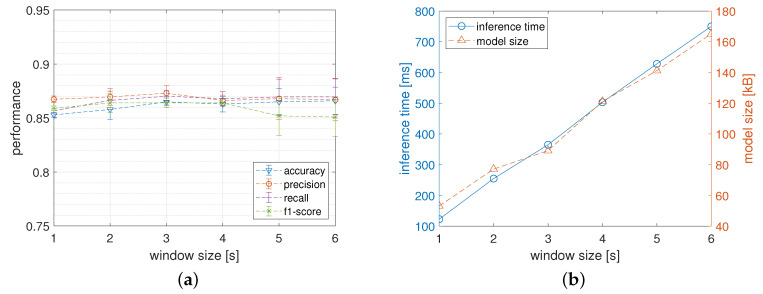
Network performance obtained when varying the size of the window processing of the $CNN\_2.2_{4}$ network. Figure (**a**) reports the classification performance while Figure (**b**) plots the memory footprint and the inference time of the model calculated on the wearable device.

**Figure 12 sensors-22-02637-f012:**
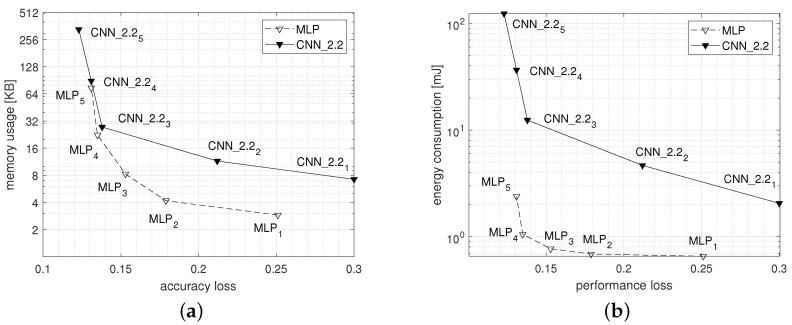
Pareto curves reporting accuracy loss Vs memory usage (**a**) and versus energy consumption (**b**) of the MLP and of the CNN_2.2 models calculated for a single inference executed on the wearable device when varying the network structure. The scale on the ordinate axis is logarithmic.

**Table 1 sensors-22-02637-t001:** SLP and MLP network structure defined in terms of the number of hidden neurons (*n*) together with the resulting trainable parameters (*p*).

*ID*	*n*	*p* (*SLP*)	*p* (*MLP*)
1	32	872	712
2	64	1736	1928
3	128	3464	5896
4	256	6920	19,976
5	512	13,832	72,712

**Table 2 sensors-22-02637-t002:** Forward feature selection results obtained with the $MLP_{4}$ network.

Features	Accuracy	Precision	Recall	f1-Score
*A*	0.761	0.758	0.758	0.755
*A + S*	0.859	**0.873**	0.868	0.869
*A + S + X*	**0.865**	**0.873**	**0.870**	**0.870**
*A + S + X + M*	0.863	0.872	**0.870**	**0.870**
*A + S + X + M + K*	0.862	0.870	0.867	0.867
*A + S + X + M + K + W*	0.859	0.871	0.866	0.867

**Table 3 sensors-22-02637-t003:** CNN structures defined in terms of number of hidden neurons (*n*) and number of convolutional filters (*j*) together with the resulting trainable parameters (*p*).

*ID*	*n*	*j*	*p* (CNN_1.1)	*p* (CNN_2.2)
1	32	4	6644	1512
2	64	8	25,824	5544
3	128	16	101,816	21,192
4	256	32	404,328	82,824
5	512	64	1,611,464	327,432

**Table 4 sensors-22-02637-t004:** Results of the McNemar test to compare MLP and CNN_2.2 performances.

	${\mathrm{MLP}}_{1}$-$CNN\_2.2_{1}$		${\mathrm{MLP}}_{2}$-$CNN\_2.2_{2}$		${\mathrm{MLP}}_{3}$-$CNN\_2.2_{3}$		${\mathrm{MLP}}_{4}$-$CNN\_2.2_{4}$		${\mathrm{MLP}}_{5}$-$CNN\_2.2_{5}$	
**Run**	**H0**	* **p** *	**H0**	* **p** *	**H0**	* **p** *	**H0**	* **p** *	**H0**	* **p** *
#1	true	$7.36\times 10^{-20}$	**false**	0.12134	**false**	0.09187	**false**	0.36484	**false**	0.26438
#2	true	$7.80\times 10^{-13}$	true	$8.41\times 10^{-18}$	true	0.00003	**false**	0.24403	**false**	0.51975
#3	true	$2.87\times 10^{-5}$	**false**	0.13615	true	0.00003	**false**	0.36831	true	0.00239
#4	true	$2.36\times 10^{-9}$	true	0.00024	**false**	0.16356	true	0.00001	**false**	0.05891
#5	true	$3.45\times 10^{-21}$	true	$5.96\times 10^{-9}$	true	0.00077	**false**	0.05528	**false**	0.10078

## Data Availability

Not applicable.

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
