# Peer review of "Exploring Artificial Neural Networks Efficiency in Tiny Wearable Devices for Human Activity Recognition"

_sensors, 2022, doi:10.3390/s22072637_

Round 1

Reviewer 1 Report

Authors conducted this research in the title of "Exploring Artificial Neural Networks Efficiency in Tiny Wearable Devices for Human Activity Recognition".

The paper’s subject could be interesting for readers of journal. Therefore, I recommend this paper for publication in this journal but before that, I have a few comments on the text that should be addressed before publication:

Comments:

1) In line 7 of Abstract section authors used this word “We”. Words like “We”, “Me”, “Our” or “Us” are not common in article writing. Other words could be used by the authors. For example, this sentence "We presented a new model in this work" could be replaced by this "A new model is presented in this work".

2) In Abstract section authors did not mention the main goal of this research. In other words, there is no obvious words about the main question that this research is managed to answer it in this section. If it is added, it would be really helpful for readers to enter and understand main purpose of this research.

3) 2. Related Work: This title "2. Related Work" in page 2 of the article should be rewrited as "Related Works". Since authors probed several works in this section, the "s" is needed after the word "Work".

4) Authors could use criteria like RMSE (Root mean squared error) or MAE (Mean absolute error) to evaluate the accuracy of conducted modelling in this article. Criteria like RMSE and MAE could be obtained based on value of real data and estimated data (achieved by the used model).

5) Figure 5 : The title of Figure 5 in page 11 of this article is too long. It should be shorter than what it is now. Authors could explain more about the figure before or after the figure. Titles should be clear and concise as much as possible.

6) What is new in this research in comparison with other similar works? In other words, authors should write more about novelty of this research. They should talk more about their innovations in this article. For example, authors should answer this question "Is the used model in this work unprecedented and completely new?". This would be useful for readers to compare this article with other similar works conducted in recent years.

7) Which software has been used in this work to export the diagrams and charts in this work?. For example, software like SigmaPlot could be utilized to export charts and diagrams.

8) Keywords: The word "Machine Learning" could be included in the Keywords part (in page 1 of this article) because it has been utilized repeatedly in different parts of this work and it seems a keyword.

9) Data Availability Statement in Conclusion section: Why authors have mentioned this phrase "Not applicable" in this part? . What are the problems about data collecting in this article?.

10) Since recently it has been proved that artificial intelligence (AI) and machine learning has a numerous applications in all of engineering fields, I highly recommend the authors to add some references in this manuscript in this regard. It would be useful for the readers of journal to get familiar with the application of AI in other engineering fields. I recommend the others to add all the following references, which are the newest references in this field

[1] Alibak, A. H., Khodarahmi, M., Fayyazsanavi, P., Alizadeh, S. M., Hadi, A. J., & Aminzadehsarikhanbeglou, E. (2022). Simulation the adsorption capacity of polyvinyl alcohol/carboxymethyl cellulose based hydrogels towards methylene blue in aqueous solutions using cascade correlation neural network (CCNN) technique. Journal of Cleaner Production, 130509.

[2] Roshani, M., et al. 2020. Application of GMDH neural network technique to improve measuring precision of a simplified photon attenuation based two-phase flowmeter. Flow Measurement and Instrumentation, 75, p.101804.

[3] Dizadji, M. R., Yousefi-Koma, A., & Gharehnazifam, Z. (2018, October). 3-Axis Attitude Control of Satellite using Adaptive Direct Fuzzy Controller. In 2018 6th RSI International Conference on Robotics and Mechatronics (IcRoM) (pp. 1-5). IEEE.

[4] Dizaji, M. R., Yazdi, M. R. H., Shirzi, M. A., & Gharehnazifam, Z. (2014, October). Fuzzy supervisory assisted impedance control to reduce collision impact. In 2014 Second RSI/ISM International Conference on Robotics and Mechatronics (ICRoM) (pp. 858-863). IEEE.

Author Response

Reviewer#1, Concern # 1: In line 7 of Abstract section authors used this word “We”. Words like “We”, “Me”, “Our” or “Us” are not common in article writing. Other words could be used by the authors. For example, this sentence "We presented a new model in this work" could be replaced by this "A new model is presented in this work".

Author response: We thank the reviewer for this suggestion. We modified the abstract as recommended. 

Reviewer#1, Concern # 2: In Abstract section authors did not mention the main goal of this research. In other words, there is no obvious words about the main question that this research is managed to answer it in this section. If it is added, it would be really helpful for readers to enter and understand main purpose of this research.

Author response: We thank the reviewer for this suggestion. We updated the abstract as recommended. 

Reviewer#1, Concern # 3: 2. Related Work: This title "2. Related Work" in page 2 of the article should be rewrited as "Related Works". Since authors probed several works in this section, the "s" is needed after the word "Work".

Author response: We thank the reviewer for this suggestion. We modified the section title as suggested. 

Reviewer#1, Concern # 4: Authors could use criteria like RMSE (Root mean squared error) or MAE (Mean absolute error) to evaluate the accuracy of conducted modelling in this article. Criteria like RMSE and MAE could be obtained based on value of real data and estimated data (achieved by the used model).

Author response: RMSE and MAE are two of the most commonly used measures for evaluating the quality of a predictor, such as a ML predictor, when the prediction is a real number continuously varying. In fact, to compute it we need to calculate the residual (difference between prediction and truth) for each data point. In the case of human activity recognition (our case) the predictions are labels such as  “walking, running, sitting, standing, etc..”  and the most appropriate way to evaluate the quality of the predictor is by computing the confusion matrix on top of which we can calculate several metrics such as Precision, Recall, F1score, and Accuracy. 

Reviewer#1, Concern # 5: Figure 5 : The title of Figure 5 in page 11 of this article is too long. It should be shorter than what it is now. Authors could explain more about the figure before or after the figure. Titles should be clear and concise as much as possible.

Author response: We thank the reviewer for this suggestion. We shortened the caption of the figure.

Reviewer#1, Concern # 6: What is new in this research in comparison with other similar works? In other words, authors should write more about novelty of this research. They should talk more about their innovations in this article. For example, authors should answer this question "Is the used model in this work unprecedented and completely new?". This would be useful for readers to compare this article with other similar works conducted in recent years.

Author response: We thank the reviewer for pointing out this suggestion. We modified the end of the Related Works section to better highlight the novelty of our study w.r.t. the current literature.

Reviewer#1, Concern # 7: Which software has been used in this work to export the diagrams and charts in this work?. For example, software like SigmaPlot could be utilized to export charts and diagrams.

Author response: Each figure reporting experimental results has been created using Matlab 2021a. 

Reviewer#1, Concern # 8: Keywords: The word "Machine Learning" could be included in the Keywords part (in page 1 of this article) because it has been utilized repeatedly in different parts of this work and it seems a keyword.

Author response: We thank the reviewer for the suggestion. We inserted it in the keywords section.

Reviewer#1, Concern # 9: Data Availability Statement in Conclusion section: Why authors have mentioned this phrase "Not applicable" in this part? . What are the problems about data collecting in this article?.

Author response: to train and test the selected artificial neural network models we used data from the realWorld HAR dataset described by Sztyler et al. in 2017 [Ref. 42 in the paper] so we have not collected any new data that may be of interest to readers.  

Reviewer#1, Concern # 10: Since recently it has been proved that artificial intelligence (AI) and machine learning has a numerous applications in all of engineering fields, I highly recommend the authors to add some references in this manuscript in this regard. It would be useful for the readers of journal to get familiar with the application of AI in other engineering fields. I recommend the others to add all the following references, which are the newest references in this field

[1] Alibak, A. H., Khodarahmi, M., Fayyazsanavi, P., Alizadeh, S. M., Hadi, A. J., & Aminzadehsarikhanbeglou, E. (2022). Simulation the adsorption capacity of polyvinyl alcohol/carboxymethyl cellulose based hydrogels towards methylene blue in aqueous solutions using cascade correlation neural network (CCNN) technique. Journal of Cleaner Production, 130509.

[2] Roshani, M., et al. 2020. Application of GMDH neural network technique to improve measuring precision of a simplified photon attenuation based two-phase flowmeter. Flow Measurement and Instrumentation, 75, p.101804.

[3] Dizadji, M. R., Yousefi-Koma, A., & Gharehnazifam, Z. (2018, October). 3-Axis Attitude Control of Satellite using Adaptive Direct Fuzzy Controller. In 2018 6th RSI International Conference on Robotics and Mechatronics (IcRoM) (pp. 1-5). IEEE.

[4] Dizaji, M. R., Yazdi, M. R. H., Shirzi, M. A., & Gharehnazifam, Z. (2014, October). Fuzzy supervisory assisted impedance control to reduce collision impact. In 2014 Second RSI/ISM International Conference on Robotics and Mechatronics (ICRoM) (pp. 858-863). IEEE.

Author response: Unfortunately, none of the suggested works has to do with machine learning applied neither in the HAR domain nor in the wearable domain. Adding these references to suggest to the reader that there are other areas in which to apply ML, we believe is useless and misleading. First of all, it would not be exhaustive of the myriad of current applications of ML, but even if it were, listing in an exhaustive way all the application domains of ML would be extremely out of place in this work which focuses on very specific aspects such as HAR and wearable devices.

Reviewer 2 Report

I appreciate the introduction of the social demand for problem solving in the introduction and also the links to the relevant literature that deals with the topic. The logic and progress of the research, as well as the results, are also described in great detail. These are presented in a detailed way, the text is suitably supplemented with figures and tables. The topic of the research is correct, the research is verifiable. The results are useful in practice and can be beneficial. I have only minor comments that I give for consideration, otherwise I evaluate the contribution very positively.

  • You do not refer to Figure 1 until page 7, but the figure itself is on page 6 at the beginning of this page. It would be better to combine the figure to appear first and then follow.
  • Research questions or hypotheses could be used and explored in research.
  • In the end, the limits of the research and proposals for further research could be evaluated in more detail.

Author Response

Reviewer#2, Concern # 1: I appreciate the introduction of the social demand for problem solving in the introduction and also the links to the relevant literature that deals with the topic. The logic and progress of the research, as well as the results, are also described in great detail. These are presented in a detailed way, the text is suitably supplemented with figures and tables. The topic of the research is correct, the research is verifiable. The results are useful in practice and can be beneficial. I have only minor comments that I give for consideration, otherwise I evaluate the contribution very positively.

Author response: thank you very much for your positive evaluation. 

Reviewer#2, Concern # 2: You do not refer to Figure 1 until page 7, but the figure itself is on page 6 at the beginning of this page. It would be better to combine the figure to appear first and then follow.

Author response: We thank the reviewer for the suggestion. We moved Figure 1 to page 7.

Reviewer#2, Concern # 3:  Research questions or hypotheses could be used and explored in research. In the end, the limits of the research and proposals for further research could be evaluated in more detail.

Author response: We thank the reviewer for the suggestion. We renamed the Conclusions section to “Conclusions, limitations and future research” and we updated the related text in order to better highlight the limitations of our study and the open future research directions.

Reviewer 3 Report

The authors analyzed the performance of multilayer perception (MLP) and convolutional neural network (CNN) on a tiny wearable device for human activity recognition applications. They claimed that it is better to use MLP than CNN to lower memory usage and energy consumption. Please see below my comments/questions.

  1. There are several machine/deep learning-based methods introduced before. The authors should propose novel algorithms and compare them with the existing works. If not possible, please provide experimental evaluations of the recent related works (e.g., https://doi.org/10.1145/2733373.2806333).
  2. A limited number of features were considered to build the models and the accuracy obtained is <90%. Is it possible to add more features in the models to improve the accuracy with reduced memory usage and energy consumption?
  3.  Please explain why MLP is showing better results than CNN. 
  4. The manuscript needs to expand the discussion on the limitations of their work and open research questions.

Author Response

Reviewer#3, Concern # 1: There are several machine/deep learning-based methods introduced before. The authors should propose novel algorithms and compare them with the existing works. If not possible, please provide experimental evaluations of the recent related works (e.g., https://doi.org/10.1145/2733373.2806333).

Author response: We thank the reviewer for pointing out this suggestion. We modified the end of the Related Works section to better highlight the novelty of our study w.r.t. the current literature.

Reviewer#3, Concern # 2: A limited number of features were considered to build the models and the accuracy obtained is <90%. Is it possible to add more features in the models to improve the accuracy with reduced memory usage and energy consumption?

Author response: we evaluate several synthetic features which are: i) average value (A); ii) standard deviation (S); iii) maximum value (X); iv) median value (M), Kurtosis (K), and Skewness (W). In Section 5.1.3. Feature selection, we report that the best results come when we use (A+S+X) and that adding more features to the selected MLP network does not further increase accuracy.  This can be explained by the fact that not all features provide original information content useful for the classification process. Moreover, calculating synthetic features entails several CPU cycles which increase the power consumption. 

Reviewer#3, Concern # 3: Please explain why MLP is showing better results than CNN.

Author response: from the only accuracy point of view CNN outperforms MLP of about 0.8 % (87.7% Vs 86.9%) but from the energy and memory usage point of view, it is more convenient to use classic Perceptron networks rather than more complex CNN. In particular, Pareto curves reporting resource utilization Vs accuracy loss show a 4x advantage of the MLP network for what concerns the memory usage, and about 36x considering the energy consumption.

Reviewer#3, Concern # 4: The manuscript needs to expand the discussion on the limitations of their work and open research questions.

Author response:  We thank the reviewer for the suggestion. We renamed the Conclusions section to “Conclusions, limitations and future research” and we updated the related text in order to better highlight the limitations of our study and the open future research directions.

Round 2

Reviewer 1 Report

All commments addressed corre tlyaccept

Reviewer 3 Report

I have no further comments. Thank you!